# Molecular Mechanisms of *Lacticaseibacillus rhamnosus*, LGG^®^ Probiotic Function

**DOI:** 10.3390/microorganisms12040794

**Published:** 2024-04-14

**Authors:** Thomas Leser, Adam Baker

**Affiliations:** Future Labs, Human Health Biosolutions, Novonesis, Kogle Alle 6, 2970 Hoersholm, Denmark; adaba@novonesis.com

**Keywords:** *Lacticaseibacillus rhamnosus*, LGG^®^, effector molecules, proteins, pili, homeostasis, tolerance

## Abstract

To advance probiotic research, a comprehensive understanding of bacterial interactions with human physiology at the molecular and cellular levels is fundamental. *Lacticaseibacillus rhamnosus* LGG^®^ is a bacterial strain that has long been recognized for its beneficial effects on human health. Probiotic effector molecules derived from LGG^®^, including secreted proteins, surface-anchored proteins, polysaccharides, and lipoteichoic acids, which interact with host physiological processes have been identified. In vitro and animal studies have revealed that specific LGG^®^ effector molecules stimulate epithelial cell survival, preserve intestinal barrier integrity, reduce oxidative stress, mitigate excessive mucosal inflammation, enhance IgA secretion, and provide long-term protection through epigenetic imprinting. Pili on the cell surface of LGG^®^ promote adhesion to the intestinal mucosa and ensure close contact to host cells. Extracellular vesicles produced by LGG^®^ recapitulate many of these effects through their cargo of effector molecules. Collectively, the effector molecules of LGG^®^ exert a significant influence on both the gut mucosa and immune system, which promotes intestinal homeostasis and immune tolerance.

## 1. Introduction

*Lacticaseibacillus rhamnosus* LGG^®^ is the most extensively clinically documented probiotic strain globally [1], boasting nearly 200 publications detailing the outcomes of randomized controlled clinical trials (RCTs). These trials have demonstrated the clinical efficacy of LGG^®^ supplementation in children and adults experiencing acute diarrhea, antibiotic-associated diarrhea, travelers’ diarrhea, pain-related functional gastrointestinal disorders, and respiratory tract infections (as reviewed by Capurso, 2019) [2], nosocomial infections [3], atopic disease [4,5,6], and as a vaccine adjuvant [7,8].

Nevertheless, achieving consistent clinical outcomes in RCTs involving LGG^®^, or any other probiotics, poses inherent challenges [2,9,10]. While RCTs have proven effective in developing precisely defined molecular pharmaceutical therapeutics for diseases, their application in other areas, such as probiotic research in healthy populations, presents a challenge. This challenge arises from the limited availability of meaningful clinical endpoints and biomarkers for probiotic function [9] and the diverse effects of probiotic effector molecules. The ambiguity is caused by our limited understanding of the molecular mechanisms governing probiotic–host interactions and the incomplete phenotypic characterization of probiotic strains. Furthermore, we do not understand the control of fundamental biological interactions between probiotics and heterogeneous resident microbiotas and how these affect probiotic function.

Additionally, probiotics, as dietary supplements or food ingredients, exert subtle effects on complex health-related conditions and biological markers, resulting in low signal-to-noise ratios when compared to targeted high-potency pharmaceutical drugs. Consequently, in RCTs involving probiotics, we are still ill-equipped to define true placebo-controlled interventions in healthy populations, and this frequently leads to ambiguous results [10].

A conventional hypothesis on probiotic functions is that they modulate the gut microbiota towards a healthier state [11,12]. While the ongoing advancements in human microbiota research have revolutionized our understanding of the importance of these symbiotic microorganisms in human health, the specific composition and functional attributes characterizing a healthy microbiota remain to be defined [13,14]. Although clinical studies provide some support for the concept of microbiota restoration in patient cohorts exhibiting dysbiosis, the reproducibility of these findings is limited [15]. Notably, in the case of healthy individuals, a systematic literature review concluded that probiotics have no discernible impact on the composition of the fecal microbiota [12]. Microbiota analyses predominantly rely on stool samples which, at best, represent the microbiota of the distal colon but not the small intestine [16]. Complicating matters further, the uniqueness of each individual’s microbiota, akin to a fingerprint, poses a significant challenge to making generalizations about probiotic effects [14,17]. These inherent complexities challenge the traditional concept of probiotic functioning through microbiota modulation, prompting a shift towards exploring mechanisms involving direct interactions between probiotic effector molecules and host physiology.

Human health transcends the mere absence of disease, and is best understood as a state of balance or equilibrium that an individual has established within himself and between himself and his social and physical environment [18]. Opposite to most drugs, the advantages of probiotics extends to the sustained maintenance of health equilibria when consumed over extended periods. By influencing fundamental physiological aspects like intestinal and immune homeostasis, as well as potentially the structure and function of the (gut) microbiome, probiotics may enhance various dimensions of an individual’s health equilibrium.

To fully realize the benefits of LGG^®^ on human health, a detailed understanding of the interactions between bacteria and human physiology at the molecular and cellular levels are needed. In vitro and animal studies are valuable tools to study these probiotic mechanisms. Understanding molecular interactions of probiotics with basic physiology transcends the confines of specific patient groups and treatment regimens and will help to recognize how LGG^®^ protects human health equilibria. This knowledge not only serves as a foundation for future research endeavors, but also facilitates the identification of valuable biomarkers and the development of pragmatic pertinent health claims to guide consumers [9,19].

In this review, we consolidate the existing knowledge regarding the molecular interactions between LGG^®^ and the host. Significant scientific contributions have been made since the review by Segers and Lebeer was published ten years ago [20]. Despite the discovery of several effector molecules within LGG^®^, we anticipate that ongoing research will unveil additional components. While RCTs have offered limited insights into probiotic mechanisms, the enduring acknowledgment of LGG^®^ as a probiotic bacterium has significantly fueled pre-clinical investigations which have created a unique understanding of the molecular mechanisms behind the health benefits of this strain.

## 2. The Digestive Tract and the Intestinal Immune System Are Targets of LGG^®^

The relation between orally ingested probiotics and the host obviously starts in the digestive tract. Through molecular interactions with mucosal and immune cells in the gastrointestinal (GI) tract, probiotics initiate signaling cascades that influence local and peripheral physiology, and eventually impacts health equilibria. Additionally, probiotics may influence the gut microbiome in some conditions, thereby causing indirect effects on human physiology.

The digestive tract encompasses the mouth, esophagus, stomach, and the small and large intestines. The intestinal epithelium is a single layer of cells that forms the lining between the body and the contents of the intestines [21]. The intestinal epithelial cells (IECs) are responsible for nutrient absorption and, at the same time, preventing noxious molecules and intestinal bacteria from entering the body. Atop these cells lies a continually replenished layer of mucus, acting as a protective barrier to keep the luminal bacteria at a safe distance from the epithelium [22,23]. IECs are the first cellular line of defense against pathogens, and react by secreting cytokines and chemokines to activate the immune system when needed [24].

Within the intestinal epithelial layer, narrow gaps create the paracellular space, sealed by intricate protein networks known as ‘tight junctions’ (TJs) to maintain a barrier between the luminal contents of the intestines and the body [25,26,27,28]. However, if TJs fail to perfectly seal these gaps for various reasons, molecular antigens such as lipopolysaccharides (LPS), or even intact bacteria, can traverse the openings, potentially triggering local inflammation [29,30]. This may be referred to as a ‘leaky gut’ [31]. Given that inflammatory molecules can circulate to other organs, the persistent dysfunction of the intestinal barrier and the failure to resolve inflammation are implicated in conditions affecting various body sites, including autoimmune, metabolic, and cardiovascular diseases [30,32,33,34,35], as well as psychological and psychiatric disorders [36,37,38,39].

Preserving the integrity of the intestinal barrier is an important feature of LGG^®^. Numerous laboratory and animal studies have consistently demonstrated the positive impact of LGG^®^ on the intestinal barrier by stabilizing TJs [40,41,42,43,44,45,46]. In humans, LGG^®^ has been shown to improve GI disorders in different age groups, and the observed positive effects on TJs and the intestinal barrier serve as a causal physiological mechanism underlying these clinically observed benefits [47,48,49].

Beneath the layer of epithelial cells lies the connective tissue, known as the lamina propria, which is intimately associated with a vast collection of immune cells referred to as the gut-associated lymphoid tissue [50]. As much as 70–80% of all immune cells in the body are located in this region [51]. The intestinal mucosa is constantly under immune surveillance to maintain tolerance towards benign microbial and dietary antigens and preserve intestinal homeostasis and, at the same time, is able to elicit appropriate immune responses to pathogens [52].

Infectious inflammatory immune responses are characterized by a quick onset of inflammation followed by a resolution and return to homeostasis [53,54]. Failure to resolve inflammation results in chronic inflammation [55], which is an underlying condition in many diseases, such as inflammatory bowel disease (IBD) [56], atherosclerosis [57], arthritis [58], and neurodegenerative diseases [59].

Amidst the diverse lymphocyte populations present in the mucosal lamina propria, CD4^+^Foxp3^+^CD25^+^ T_reg_ cells (regulatory T cells) are essential for maintaining immune homeostasis [60]. They contribute to preserving peripheral tolerance, preventing autoimmune diseases, and moderating inflammation induced by pathogens and environmental insults [61,62]. In healthy individuals, the intestinal mucosal immune system is tightly regulated, whereas inadequate regulation may cause intestinal inflammation and tissue damage as observed in patients with IBD [62].

A hallmark of LGG^®^ interaction with the immune system is the expansion of regulatory T cells. In vitro and animal studies have consistently demonstrated that LGG^®^, by inducing the expression of transforming growth factor beta (TGF-β) expression in IECs, promotes the differentiation of T cells into CD4^+^Foxp3^+^CD25^+^ T_reg_ cells which regulate intestinal homeostasis [63,64,65,66,67,68,69].

Maintaining the intestinal epithelium integrity and immune homeostasis are fundamental physiological functions that govern human health and well-being. Disruption of these functions not only has local implications, but also extends to peripheral structures such as the central nervous system, the cardiovascular system, metabolism, and systemic immunity.

## 3. LGG^®^ Communicates with the Human Host through Molecular Mechanisms

Probiotics may interact with the host throughout the GI tract, but physiologically important interactions are more likely to occur in the small intestine than in the colon due to the lower bacterial densities and the anatomical and functional characteristics inherent to the small intestine.

Cell surface molecules and secreted molecules from probiotic bacteria are mediators of bacteria–host signaling and the health benefits of these bacteria [70]. Molecules situated on the bacterial cell surface necessitate direct bacteria–cell contact to exert their effects [71]. The mucus layer lining the intestinal epithelium typically excludes such interactions, except in specific areas of the mucosa like Peyer’s patches (PPs) where the immune cells sample intestinal contents. However, it has recently been discovered that bacteria also communicate with the human host through secreted extracellular vesicles (EVs), which can traverse the mucus layer. In contrast to surface-anchored molecules, secreted compounds do not require direct bacteria–cell contact for signaling; instead, they can remain in solution and penetrate the mucus layer to interact with underlying epithelial cells. Moreover, these secreted molecules may be absorbed by the mucosa and disseminated to peripheral organs through the blood or lymphatic circulation.

Several surface anchored molecules which affect the intestinal mucosa and immune system, including the pili, peptidoglycan, lipoteichoic acid, exopolysaccharides, as well as the secreted proteins p40, p75, and HM0539 have been identified in LGG^®^. These effector molecules, despite sometimes inducing seemingly opposing immune responses, collectively contribute to the overall enhancement of immune regulation and protection of the intestinal mucosa by LGG^®^. Thus, LGG^®^ has been shown to inhibit cytokine-induced apoptosis, preserve barrier function, up-regulate mucus production, promote immune tolerance and homeostasis, and stimulate immunoglobulin A (IgA) production which protects against intestinal inflammation.

A serious challenge in studying effector molecules lies in the technical complexities associated with purifying certain molecular components. Ideally, only pure molecular fractions should be studied. However, extracting pure molecular fractions, uncontaminated by other substances, can be challenging. To further complicate this, the bacterial synthesis of structural and secreted compounds may be sensitive to fermentation conditions such as specific nutrient availability or temperature [72,73,74]. Consequently, findings from such studies should be approached with caution, acknowledging the potential influence of growth conditions and impurities on the results.

## 4. Secreted Proteins from LGG^®^ Have Probiotic Effects

The p75 (Major secreted protein 1 (Msp1)) and p40 (Major secreted protein 2 (Msp2)) proteins are the two most abundantly secreted proteins found in spent culture supernatant of LGG^®^ [75]. These proteins are also present as cell surface layer proteins [75] on extracellular vesicles [76], and in LGG^®^-fermented milk [77]. Both proteins are cell wall muramidases, which are essential for a normal separation of LGG^®^ cells during growth [75]. p40 has 412 amino acid residues with a 42 kDa molecular mass [75]. Secreted p75 is an *O*-glycosylated 498 amino acid protein with a molecular mass of 75 kDa [78]. p40 and p75 are not unique to LGG^®^. True p40 proteins were found in strains of the *L. casei* group (includes *L. rhamnosus*), and in the *Latilactobacillus sakei* and *Ligilactobacillus salivarius* groups. p40 homologs were found in species of the *Enterococcaceae* and *Streptococcaceae* families, suggesting that the p40 encoding gene was present in a common ancestor. p75-related proteins were only found in the *L. casei* and *L. sakei* groups [79].

Both in vitro and animal studies have demonstrated the signaling capabilities of p40 and p75 to host cells. Comparisons have revealed that p40 exerts several folds stronger effects on host cells than p75 [80]. The functional domain of p40, located at the N-terminal 1–180 amino acid peptide, exhibits robust signaling to intestinal IECs comparable to the full-length p40 [80]. Several studies have shown that p40 activates the epidermal growth factor receptor (EGFR) [69,77,80,81,82,83,84,85]. The activation of EGFR leads to the stimulation of intracellular signaling networks, triggering biological responses, such as reduced apoptosis, enhanced cell survival, increased mucus production, fortification of cellular tight and adherens junctions, and immune regulation. Collectively, these effects contribute to the maintenance of intestinal epithelial homeostasis.

Intriguingly, colonic epithelial cell-derived components have been shown to promote p40 protein synthesis and secretion by LGG^®^, thereby enhancing the protective effects of LGG^®^ on IECs. Extracellular vesicles secreted by human cells, containing heat shock protein 90 (Hsp90), play a role in mediating communication with LGG^®^ in this context [86]. This finding is an example of the intricate interplay between IECs and LGG^®^, which ultimately protects the integrity of the intestinal epithelium and prevents intestinal inflammation.

### 4.1. p40 and p75 Improve Intestinal Epithelial Integrity

p40 does not directly bind to and activate EGFR; rather, its activation of a disintegrin and metalloproteinase domain-containing protein 17 (ADAM17) is crucial for EGFR activation. This, in turn, leads to the release of the EGFR ligand, heparin-binding, EGF-like growth factor (HB-EGF) [82]. The subsequent stimulation of EGFR by HB-EGF initiates several downstream signal transduction cascades, such as phosphoinositide-3-kinase–protein kinase B/Akt (PI3k/Akt) signaling, mitogen-activated protein kinases (MAPK), and c-Jun N-terminal kinases (JNKs), which promote DNA synthesis and cell proliferation and survival.

In vitro studies have demonstrated that p40’s activation of EGFR triggers the PI3k/Akt signaling pathway, resulting in reduced cytokine-induced programmed cell death (apoptosis) and the increased survival of IECs [82,87,88]. Mouse colitis models further support these findings, showing that p40-stimulated PI3k/Akt signaling reduces colon epithelial cell apoptosis, preserves epithelial barrier function, and mitigates chronic inflammation [69,77,81,82]. The administration of LGG^®^ and p40 to neonatal mice has been associated with the promotion of intestinal maturation through EGFR-mediated PI3k/Akt activation in IECs [43,69]. Neonatal mice, orally administered p40-containing hydrogels, exhibited a significant increase in body weight from postnatal day 8 to 20 compared to control hydrogels, with no discernible differences observed in mice older than 3 weeks. In 2-week-old mice, p40 supplementation increased IEC proliferation in both the small intestine and colon, but not in older mice. In 3 week-old mice, p40 increased gene expression and localization of digestive enzymes, tight junction proteins, and mucus gene expression. These mice had increased resilience to intestinal injury and colitis later in life [69].

Furthermore, the LGG^®^-conditioned medium has shown dose–dependent promotion of the growth of colonoids and prevention of tumor necrosis factor alpha (TNF-α)-induced cell apoptosis. The enhanced recovery of colonoids from cytokine-induced injury was attributed to the inhibition of apoptosis and increased IEC proliferation from intestinal stem cells stimulated by the conditioned medium. These effects were attributed to unidentified, secreted molecules present in the LGG^®^ supernatant [89].

The mucus layer lining the gastrointestinal tract is an important defense mechanism of the intestinal barrier and, as such, contributes to intestinal epithelial homeostasis. p40-stimulated EGFR activation and PI3k/Akt signaling has been shown to up-regulate mucin 2 (MUC2) gene expression and mucin production in a concentration-dependent manner in vitro, and to promote mucin production in the colonic epithelium in mice [69,83]. In humans with type two diabetes mellitus, an 8-week consumption of LGG^®^ led to a remarkable over ninefold increase in fecal mucin 2 and mucin 3A gene expression compared to the baseline, while no such changes were observed in the placebo group [90]. Although the molecular mechanisms involved were not identified, it is plausible that the effect was caused by LGG^®^-secreted proteins such as p40, or HM0539 (see below).

p40 and p75 also protect the intestinal epithelium from oxidative stress. When applied to IECs in vitro, p40 and p75 treatment mitigated H_2_O_2_-induced disruption of TJs and adherens junctions. This protection was achieved through the EGFR-regulated activation of protein kinase C and extracellular-signal-regulated kinase (ERK), thereby preventing the redistribution of occludin (OCLN), zonula occludens-1 (ZO-1), E-cadherin, and β-catenin from intercellular junctions to the intracellular compartment [85]. In neonatal mice, p40 promoted the formation of TJs formation in IECs by stimulating the expression of the TJ protein claudin-3 and enhancing the membrane localization of ZO-1 by activating EGFR [69]. Additionally, the nuclear factor erythroid 2-related factor 2 (Nrf2)-tight junction signaling pathway may be involved in the p40 stimulation/preservation of TJs (see paragraph on Extracellular Vesicles).

Pre-treatment of human enteroids and colonoids with the LGG^®^-conditioned medium (CM) prevented interferon-γ (IFN-γ)-induced increases in paracellular permeability and the down-regulation of OCLN and ZO-1 gene expression. While the specific effector molecules in CM were not identified, it is likely that proteins were involved, as boiled CM, LGG^®^ cell walls, or extracted DNA had no impact on barrier function. Although CM activated EGFR and ERK signaling, this mechanism alone did not account for the barrier-protecting effect, suggesting the involvement of other signaling pathways evoked by LGG^®^ CM [91].

Protein fractions from the LGG^®^-conditioned medium were shown to induce expression of heat shock proteins (Hsp25 and Hsp72) in IECs. Heat shock proteins are cytoprotective and increase cellular tolerance towards thermal, osmotic, oxidative, and inflammatory stress, and thus, fortify the epithelial barrier against damage from a variety of injurious insults. Notably, heat shock protein-encoding genes exhibited the most robust up-regulation in colonic epithelial cells following treatment with the LGG^®^-conditioned medium. The induction of heat shock proteins was mediated through p38 mitogen-activated protein kinases (p38/MAPK) and JNK signaling pathways [92]. While the specific LGG^®^ proteins responsible for these effects were not conclusively identified, they are presumed to be secreted acid- and heat-stable low molecular weight peptides [88].

### 4.2. p40 Is Immune Regulatory

p40 modulates various aspects of immunity, potentially contributing to overall homeostasis. Thus, in colitis mouse models, p40 reduced intestinal production of pro-inflammatory cytokines, including TNF-α, interleukin (IL) 6, keratinocyte chemoattractant (KC), and IFN-γ [80,81]. In human-derived peripheral blood mononuclear cells (PBMCs), live LGG^®^ induced both pro- and anti-inflammatory cytokines with a bias toward an anti-inflammatory response evidenced by stronger stimulation of IL-10 and TGF-β secretion. Heat-killed LGG^®^ yielded lower cytokine secretion from PBMCs, suggesting a potential contribution of bacterial metabolites to cytokine responses [93]. In atopic children, the consumption of LGG^®^ significantly increased serum concentrations of IL-10 [94]. Although not explicitly linked to any effector molecules, p40 is as a likely candidate responsible for the immune-regulatory effects observed in these studies.

Promotion of intestinal IgA production and abundance of IgA expressing B cells within the lamina propria of the small intestine by p40 have been found [69,84]. Again, this effect is mediated by EFGR activation and PI3k/Akt signaling, leading to the up-regulation of a proliferation-inducing ligand (APRIL) expression in IECs. APRIL, in turn, triggers B cell IgA class switching in the lamina propria, resulting in increased IgA production. B cells treated directly with p40 did not increase IgA production [84]. Mice administered p40 in early life retained elevated fecal IgA levels in adulthood and had reduced susceptibility to intestinal injury and dextran sulfate sodium (DSS)-induced colitis [69].

T_reg_ cells are crucial for immune tolerance and homeostasis. The dysregulation of T_reg_ cell function invariantly disrupts immune homeostasis and tolerance and often causes autoimmunity [60]. The TGF-β superfamily plays a central role in immune homeostasis by controlling T_reg_ cell generation and function through intricate, context-dependent mechanisms [60]. Notably, p40 has been shown to stimulate the secretion of TGF-β from intestinal cells in various settings, including cell cultures, enteroids, colonoids, and murine models [63,69]. In neonatal mice, p40 treatment significantly amplified the abundance of T_reg_ cells in the lamina propria of the small intestine by enhancing TGF-β production in IECs [69].

Moreover, a compelling mechanism of p40-induced epigenetic imprinting has been elucidated in mice. Early life supplementation of p40 or a recombinant peptide, p40N120, (28–120 residues), increased the expression of a methyltransferase (su(var)3-9, enhancer-of-zeste, and trithorax domain–containing 1b) (*Setd1b*) in IECs, resulting in histone H3 methylation (H3K4me1/3) and, consequently, elevated gene expression of TGF-β. This enhanced TGF-β production led to the expansion of T_reg_ cells, conferring protection against epithelial disruption and inflammation [63,95]. Interestingly, early exposure to p40 had persistent protective effects even in adulthood, probably through the epigenetic reprogramming of intestinal stem cells. Breast milk, a primary source of TGF-β, exerts functional effects in neonates [96]. The supplementation of LGG^®^ to pregnant and lactating mothers has been associated with increased concentrations of TGF-β2 in mother’s milk, thereby enhancing the immunoprotective potential of breast milk. This augmented TGF-β2 content was correlated with a reduced risk of children developing atopic eczema during the first 2 years of life [4]. The synergistic action of TGF-β and IL-10 in breast milk may play a preventive role against sensitization to food allergens by promoting specific IgA production and expanding T_reg_ cells [96].

### 4.3. HM0539: A Secreted Protein with Probiotic Effects

Recently, another major secreted protein from LGG^®^ with probiotic effects was discovered. This protein, provisionally named HM0539 [41], also protects the intestinal barrier and attenuates inflammatory responses. Both in vitro and animal studies have indicated that HM0539 effectively prevents barrier injury induced by lipopolysaccharide (LPS) and TNF-α. Its protective mechanisms involve the stimulation of mucin production in IECs and the preservation of gene expression and localization of TJ proteins (ZO-1, OCLN), thereby maintaining the integrity of the intestinal barrier [41]. Furthermore, HM0539 has anti-inflammatory properties. In experiments involving LPS-stimulated macrophages and murine colitis induced by DSS, HM0539 reduced the pro-inflammatory activation of toll-like receptor (TLR) 4 and myeloid differentiation primary response 88 (MyD88)-dependent signaling. This, in turn, attenuated nuclear factor kappa B (NF-κB)-dependent inflammatory responses [97].

## 5. Extracellular Vesicles

As part of their normal life cycle, bacteria produce extracellular vesicles (EVs). EVs are outer membrane lipid bilayer nanostructures, ranging from 20 to 300 nm in size, which are secreted extracellularly. They contain cargoes of proteins, lipids, peptidoglycans, LPS, and nucleic acids. EVs allow for the long-distance delivery of active bacterial compounds in a protective environment and are important means of communication between prokaryotic and eukaryotic cells [98]. EVs are resistant to enzymatic degradation and acid and can interact with and cross the intestinal epithelium and access underlying immune cells. They have also been found in the bloodstream from where they have access to peripheral tissues, including the brain [99]. Although still in the early stages of research, in vitro and animal studies have shown that EVs are able to recapitulate many of the probiotic effects exhibited by the bacterial strains they originate from and are highly important elements in probiotic mechanisms of action [100].

The secreted p40 and p75 proteins are abundantly associated with LGG^®^ extracellular vesicles [76]. In the *L. casei* BL23-conditioned medium, cell wall muramidases akin to p40 and p75 have been identified. All of the p40 and most of the p75 found in the medium were associated with EVs. The two proteins were bound to the surface of BL23 EVs, possibly by binding to lipoteichoic acid embedded in the phospholipid bilayer of EVs. BL23 EVs activated phosphorylation of EGFR in a dose-dependent manner similar to the p40 and p75 purified proteins [101].

LGG^®^-derived EVs have been shown to prevent colonic tissue damage and shortening of the colon in the DSS colitis mouse model. EVs ameliorated DSS-induced colonic inflammation by inhibiting TLR4/NF-κB/NLRP3 (NOD-, LRR- and pyrin domain-containing protein 3) inflammasome activation. Consequently, reduced serum levels of pro-inflammatory cytokines (TNF-α, IL-1β, IL-6, IL-2) were measured in EV-treated mice. Additionally, modifications of the gut microbiota were seen [102]. In IECs, LGG^®^-derived EVs strengthened the intestinal barrier by enhancing the expression of TJ proteins, including ZO-1, OCLN, and claudin-1. Treatment of macrophages with EVs protected from LPS, or *Staphylococcus aureus*-induced inflammatory responses by suppressing the expression of pro-inflammatory cytokines (TNF-α, IL-1β, IL-6, monocyte chemoattractant protein 1 (Mcp-1), IFN-γ, and IL-17A) [76,103].

In a murine model of alcohol-associated liver disease (ALD), EVs derived from LGG^®^ protected against alcohol-induced dysfunction of the intestinal barrier and liver steatosis. Alcohol feeding decreased the ileal expression of ZO-1, OCLN, and claudin-1, accompanied by an elevation in serum LPS levels. However, treatment with LGG^®^ EVs prevented the down-regulation of these proteins and reduced circulating LPS levels. Notably, LGG^®^ EVs contained high levels of p75 and p40. EVs were shown to increase intestinal IL-22/regenerating islet-derived protein 3 (Reg3) and Nrf2-tight junction signaling pathways, effectively inhibiting bacterial and LPS translocation in ALD mice. This protective effect was, at least partially, attributed to the activation of the aryl hydrocarbon receptor (AhR) signaling pathway, stimulated by the bacterial tryptophan metabolites (indole-3-aldehyde, indoleacrylic acid; and indole-3-lactic acid) contained in EVs [76].

In hepatic cancer cells (HepG2), LGG^®^ Evs had a cytotoxic effect as determined by a significantly increased apoptotic index (apoptosis regulator (BAX)/B-cell lymphoma (Bcl-2) expression ratio) that led to cancer cell death [104]. A cytotoxic effect, coupled with reduced cell proliferation was also observed in the HT29 and SW480 colorectal cell-lines following treatment with LGG^®^ Evs [105].

## 6. LGG^®^ Surface Molecules with Probiotic Effects

### 6.1. Pili and Other Proteinaceous Adhesins

The cell surface of LGG^®^ is covered with long proteinaceous fimbria-like appendages known as pili. On average, 10 to 50 pili are present on each cell, predominantly located near the cell poles. LGG^®^ pili are encoded by the *SpaCBA* gene cluster [106]. The SpaCBA pili are key to LGG^®^ biofilm formation and to the broad binding specificity to mucins, extracellular matrix and IECs [107,108]. The SpaC pilin, positioned at the pilus tip and extending along its length, confers a high binding capacity, particularly to β-galactoside-containing carbohydrate moieties present in intestinal mucus [106,109]. The interaction between LGG^®^ and host carbohydrates initiates with the SpaC subunit at the pilus tip, followed by a more intricate contact facilitated in a zipper-like fashion by dispersed SpaC subunits along the pilus length. Additionally, SpaC promotes binding to SpaC on other LGG^®^ cells, causing bacterial aggregation and facilitating biofilm formation [108].

Two additional surface proteins contributing to LGG^®^ mucosal adhesion have been identified, although their role in attachment is likely less important than the SpaCBA pili. The modulator of adhesion and biofilm (MabA) modulates LGG^®^ adhesion to IECs and biofilm formation. However, LGG^®^ mutants lacking MabA expression are still able to adhere to Caco-2 cells and form biofilms, albeit with a reduced capacity [110]. The mucus-binding factor (MBF) is a mucus-specific, cell wall-associated adhesin, presumably involved in pilus-mediated mucosal adhesion [111]. It is hypothesized that the initiation of mucosal adhesion involves the SpaC subunit on LGG^®^ pili, with subsequent establishment of more stable interactions between bacteria and mucosa facilitated by MabA and MBF [110,111].

In addition to their adhesive role, pili contribute to the interactions between LGG^®^ and the immune system. SpaCBA pili undergo post-translational mannose and fucose glycosylation, and these glycans are recognized by DC-SIGN, a crucial pattern recognition receptor on human dendritic cells (DCs) [112]. Purified glycosylated pili from LGG^®^ were found to induce gene expression of pro-inflammatory cytokines (IL-6, IL-12), as well as immune regulatory IL-10 expression in immature human DCs [112]. Interleukin 8 is a chemoattractant cytokine produced by IECs which recruits and activates neutrophils in inflammatory regions. LGG^®^ pili have been shown to attenuate IL-8 expression in IECs triggered by other cell surface components of LGG^®^, such as LTA [107].

Pilus-mediated adhesion to macrophages (RAW 264.7 cells), leading to enhanced bacteria-to-cell contact, has been associated with an anti-inflammatory response characterized by the induction of IL-10 gene expression, concomitant with a reduction in pro-inflammatory IL-6 gene expression [71]. The presence of pili also stimulated macrophage phagocytosis of LGG^®^. Although the specific mechanisms underlying this immune modulation remain unclear, it is plausible that the SpaCBA pili, by facilitating close bacteria-macrophage connections, enable more intimate interactions with other potent LGG^®^ effector molecules [20].

In vitro, LGG^®^ lysate was shown to inhibit *S. aureus* adhesion to epidermal keratinocytes and to increase the rate of re-epithelialization of injured keratinocyte monolayers [113]. SpaC was involved in this function, as recombinant SpaC inhibited *S. aureus* adhesion to keratinocytes in a dose-dependent manner and improved keratinocyte viability following *S. aureus* challenge. However, enolase and triosephosphate isomerase present in the LGG^®^ lysate were likely also involved in these effects [114].

The SpaCBA pili of LGG^®^ have considerable sequence homology and structural relation to the PilB-type pili of enterococci. PilB-type pili are implicated in the binding of vancomycin-resistant *Enterococcus faecium* to intestinal mucus and subsequent colonization of the human GI tract. In a mechanism of competitive exclusion, purified SpaC pilin from LGG^®^ was able to significantly reduce the binding of *E. faecium* to mucus and may, thus, have potential as a new treatment strategy against vancomycin-resistant enterococci [115].

### 6.2. Lectin-like Cell Wall Proteins

Two cell wall proteins with N-terminal Legume-type (L-type) lectin domains have been identified in LGG^®^—namely, (putative lectin-like protein 1 and 2 (Llp1 and Llp2)). Lectins are carbohydrate-binding proteins that mediate cell interactions, including binding and attachment. The lectin domain of Llp1 was shown to selectively bind to mannan polymers, while the Llp2 lectin domain was bound to both mannan and D-mannose. Both lectins recognize and bind to complex glycan structures on IECs, in vitro, suggesting a role in the adhesion capacity of LGG^®^. The lectin domains of Llp1 and Llp2 were found to inhibit bacterial biofilm formation and disrupt existing biofilms, particularly against clinical isolates of *Salmonella* Typhimurium and uropathogenic *Escherichia coli*. Llp2 displayed a somewhat stronger effect in this context [116].

### 6.3. Exopolysaccharides

Like other lactobacilli, LGG^®^ produces exopolysaccharides (EPS). EPS are water-soluble long-chain polysaccharides featuring branched, repeating units of sugars or sugar derivatives [117]. Two major types of cell wall-associated polysaccharides have been identified on the LGG^®^ cell surface, i.e., long galactose-rich polysaccharides and shorter glucose/mannose-rich polysaccharides [20]. The genomic characterization of LGG^®^ has unveiled a distinct and strain-specific EPS gene cluster responsible for the synthesis of the galactose-rich polysaccharides. A mutant strain, incapable of producing the long galactose-rich polysaccharides, was found to adhere better to mucus and IECs than the wild-type LGG^®^. This was attributed to the absence of the EPS layer, which otherwise shields adhesins, such as pili [118]. The density of LGG^®^ EPS is dependent on pH. At pH 6.8, the EPS layer is soft, and pili are easily accessible for binding, while at pH 4.8, the EPS matrix becomes compacted, embedding pili within and limiting their binding capacity due to steric hindrance [119]. Accordingly, stomach acid is likely to diminish LGG^®^’s binding capacity to intestinal mucus and the extracellular matrix. Conversely, exposure to bile salts induces a looser and thicker EPS layer, facilitating pili exposure and enhancing LGG^®^ binding strength to mucus and the extracellular matrix [120]. Moreover, exposure of LGG^®^ to bile salts triggers a reduction in EPS biosynthesis as part of a major stress response [121].

EPS protects LGG^®^ against certain adverse conditions in the GI tract and promotes survival and persistence by forming a shield against antimicrobial peptides and complement factors. As another example of the intricate relationship between IECs and LGG^®^, it has been observed that sub-inhibitory concentrations of antimicrobial peptides stimulate EPS production in LGG^®^, fortifying the defense mechanisms of the bacterial cells [122].

In vitro studies revealed that LGG^®^, but not an EPS-deficient mutant, was able to interfere with the basic physiological mechanisms of *Candida albicans* pathogenesis. Both wild-type LGG^®^ and purified EPS inhibited *C. albicans* adhesion, hyphal morphogenesis, and tissue invasion in human vaginal and lung epithelial cell lines [123]. However, EPS may not be the most important LGG^®^-derived inhibitor of *C. albicans*. It was later shown that EPS only inhibits hyphal morphogenesis at rather high concentrations, whereas the p75 secreted protein is very inhibitory, even at low concentrations. Additionally, lactic acid inhibited hyphal growth and a synergistic effect of p75 and lactic acid was found. The anti-hyphal effect of p75 was demonstrated to be due to chitinase activity of the protein which degrades the chitin polymers in the hyphal cell wall of *C. albicans* [124].

Protection of the intestinal epithelium by purified LGG^®^ EPS through anti-inflammatory and anti-oxidative effects have been demonstrated. In IEC cultures, LGG^®^ EPS reduced LPS-induced inflammatory cytokines by inhibiting p38/MAPK and NF-κB signaling pathways [125]. In a murine model infected with *Salmonella* Typhimurium, EPS from LGG^®^ alleviated *Salmonella*-induced intestinal injury by regulating TLR4/NF-κB/MAPK signaling in a dose dependent manner. EPS significantly reduced serum levels of pro-inflammatory cytokines induced by *Salmonella* (IL-1β, IL-2, IL-6 and TNF-α), mitigated *Salmonella*-induced oxidative damage in the ileum, and improved weight loss of infected mice. These effects were not due to any direct bactericidal effect of EPS against *Salmonella* [126].

Reactive oxygen species (ROS), such as ^•^O_2_^−^ (superoxide), H_2_O_2_ (hydrogen peroxide), and ^•^OH (hydroxyl radicals), are inherent byproducts of cellular oxygen metabolism maintained at low and steady levels within cells for essential physiological functions. However, an imbalance between ROS production and the efficiency of the antioxidant system can lead to the accumulation of ROS, causing oxidative stress in cells and tissues [127,128]. Anti-oxidative effects of purified LGG^®^ EPS through ROS scavenging and stimulation of cellular anti-oxidative enzymes have been found, in vitro. When IECs were exposed to H_2_O_2_, EPS from LGG^®^ effectively reduced ROS release while enhancing the activities of anti-oxidative enzymes such as superoxide dismutase (SOD) and glutathione peroxidase (GSH-Px). A consequence of oxidative stress in intestinal tissues is damage to TJs and cell apoptosis, causing intestinal barrier dysfunction [128]. LGG^®^ EPS was found to preserve TJs by stimulating the expression of ZO-1, OCLN and claudin-1, and to alleviate cell apoptosis induced by H_2_O_2_ via the Nrf2-antioxidant response element signaling pathway [129].

EPS from LGG^®^ may potentially affect lipid metabolism as evidenced by reduced adipogenesis in EPS treated fibroblast cells and improved lipid metabolism in mice subjected to a high-fat diet (HFD). In vitro, EPS attenuated triacylglycerol (TAG) accumulation in adipocytes by suppressing lipogenesis through a TLR2-dependent mechanism. This suppression was marked by a significant reduction in the expression of key adipogenesis markers, while markers of lipolysis remained unaffected. When administered intraperitoneally to mice on an HFD, LGG^®^ EPS decreased the TAG levels in serum and liver, reduced fat pads, diminished adipocyte size, and alleviated inflammation, suggesting an improved lipid metabolism in the HFD-mice [130].

### 6.4. Lipoteichoic Acids

Lipoteichoic acids (LTAs) are polymers and major constituents of the cell wall of Gram-positive bacteria. LTAs contribute to cell morphology, adhesion, biofilm formation, survival under hostile conditions and interaction with the immune system [117]. The structural configuration and net anionic charge of LTA are determined by D-alanyl ester substitutions, contributing to the molecular characteristics of the cell surface of LGG^®^. D-alanylation of LTA requires proteins encoded by the *dlt* operon, and a *dltD* mutant of LGG^®^ was completely devoid of LTA D-alanyl esters. Several characteristics of LGG^®^ were severely modified in the *dltD* mutant, including cell morphology, acid tolerance, and sensitivity to antimicrobial agents [131].

In general, LTAs are immunomodulatory molecules interacting with TLR2/6 triggering activation of NF-κB and subsequent transcription of pro-inflammatory cytokines and chemokines [132]. Such immune responses have also been detected when DCs were exposed to LTA purified from LGG^®^. LTA induces immune responses analogous to LPS, but the stimulatory potency of LTA is considerably lower than that of LPS [133]. In IEC cultures, purified LTA from LGG^®^ activated TLR2/6 and NF-κB signaling, leading to the secretion of pro-inflammatory IL-8 [134]. In the DSS-colitis mouse model, LGG^®^ had no effect, or even exacerbated colitis symptoms. Conversely, the *dltD* mutant ameliorated colitis by down-regulating TLR2 and downstream pro-inflammatory cytokine expression, suggesting that the pro-inflammatory nature of LGG^®^ LTA is attributed to D-alanyl ester substitutions [135]. In apparent contrast, LTA from LGG^®^ suppressed *Salmonella* Typhimurium flagellin-induced IL-8 secretion from porcine peripheral blood mononuclear cells, thus displaying an anti-inflammatory profile [136].

The small intestinal epithelium is highly sensitive to radiation and is a major site of injury during radiation therapy and environmental overexposure. LGG^®^, or its conditioned culture medium, reduced radiation-induced epithelial injury and improved crypt survival in mice [137]. This radio-protective effect of LGG^®^ was mediated by LTA activation of macrophages and the migration of cyclooxygenase-2 (COX-2) expressing mesenchymal stem cells (MSCs) to areas adjacent to epithelial stem cell niches. Prostaglandin E2 (PGE2), secreted by MSCs, protected the epithelial stem cells from radiation-induced apoptosis by a mechanism similar to the wound repair process [138]. PGE2 was also involved in LGG^®^ protection of the gastric mucosa in rats from ethanol-induced damage. LGG^®^ pre-treatment of rats significantly elevated basal gastric mucosal PGE2 levels, resulting in increased mucus thickness, reduced cell apoptosis, and reduced susceptibility to alcohol-induced damage [139].

UV radiation is a major cause of skin cancer, instigating epidermal hyperplasia and immunosuppression involving regulatory T-cells [140]. In UVB-induced skin cancer mouse models, the oral administration of LTA from LGG^®^ had both prophylactic and therapeutic anti-tumoral effects. LTA modulated both the innate and the adaptive skin immune systems, countering UVB-induced immunosuppression. This resulted in the proper recruitment of monocytes to the inflamed skin and facilitated adequate T cell activation and effector function [140,141].

Orally administered LTA from LGG^®^ can increase the number of IgA-positive B cells in the small intestinal lamina propria and elevate fecal IgA contents in mice [140,142]. In bone-marrow derived DCs, LTA up-regulated expression of IL-6, retinaldehyde dehydrogenase 2 (RALDH2), and IL-10. DCs expressing RALDH metabolize retinol into all-*trans* retinoic acid (RA), which together with IL-10 promotes T_reg_ cell responses that regulates the affinity maturation of IgA in Peyer’s patches. DCs expressing IL-6 enhance IgA production from B cells via IL-6R signaling [142].

IEC-derived components have recently been shown to change the transcriptional profile of LGG^®^, with a particular impact on pathways associated with nutrient acquisition and LGG^®^ growth. The conditioned cell medium from IECs led to an up-regulation of genes associated with LTA biosynthesis, including various D-alanyl-LTA biosynthesis proteins, again emphasizing the intimate and regulated nature of the interactions between LGG^®^ and the human host [143].

### 6.5. Peptidoglycan

Lactobacilli have a cell wall with a thick peptidoglycan (PG) layer that form a rigid structure for protecting the bacterial cells. PG, or murein, are polysaccharides cross-linked via peptide bridges whose detailed structure varies between bacterial species and strains of the same species [144]. The major secreted proteins, p40 and p75 are cell wall muramidases that release peptidoglycan fragments [75]. PG and PG fragments are recognized by pattern recognition receptors like TLR2, NOD-like receptors, or peptidoglycan recognition proteins (PGlyRPs) expressed in IECs [144,145].

Interactions between LGG^®^ peptidoglycan or its fragments and the human host remain a subject that warrants further exploration. In Caco-2 intestinal epithelial cells, PG from LGG^®^ elicited an acute up-regulation of the expression of pro-inflammatory cytokines (IL-12p35, IL-8 and TNF-α) mediated via TLR2 and MyD88 signaling. Notably, with a time-delay, LGG^®^ PG also stimulated Peptidoglycan Recognition Protein 3 (PGlyRP3) gene expression, exerting an anti-inflammatory effect evidenced by the subsequent reduction in pro-inflammatory cytokine expression following PG stimulation [146].

## 7. LGG^®^ Genomic DNA and Unmethylated CpG-Rich DNA Motifs

Bacterial DNA can stimulate mammalian immune cells. Stimulation depends on unmethylated cytosine-guanine (CpG) oligodeoxynucleotides (ODN) motifs present in bacterial DNA that are released upon bacterial lysis. In macrophages and DCs, CpG DNA activates intracellular TLR9 and induce a T-helper 1 (T_h1_) immune response through MyD88/NF-κB signaling [147]. A potent ODN (ID35) with a TTTCGTTT motif was identified in the LGG^®^ genome. Experimental findings in murine splenocytes revealed that ID35 induced B cell proliferation, activated DCs, and triggered the expression of Th1-type pro-inflammatory cytokines (IL-6, IFN-γ, TNF-α, IL-12p35, and IL-18) gene expression. B cell proliferation upon ID35 treatment was also seen in human PBMC-derived B cells [148]. Furthermore, the ingestion of a single high dose of LGG^®^ (8.85 × 10^11^ CFU) led to transcriptional up-regulation of genes associated with B cell activation in jejunal biopsies after 2 h in 1/3 of healthy human volunteers. No consistent changes of transcription profiles were detected in the remaining volunteers, potentially due to a delayed LGG^®^ response in these individuals [149].

The T_h1_-skewing impact of ID35 could be linked to the favorable outcomes of LGG^®^ in allergic diseases [5,20,150,151], provided that an adequate number of LGG^®^ cells undergo lysis post-ingestion to release substantial amounts of DNA motifs. In an ovalbumin-sensitized mouse model, synthetic ID35 was a potent suppressor of antigen-specific immunoglobulin E (IgE) production. The mechanism involved the activation of the CD11c^+^CD8a^+^ subset of dendritic cells associated with T_h1_ systemic responses, induction of IFN-γ production by CD4^+^ T cells, and a decrease in serum levels of total and antigen-specific IgE [152].

Unlike the pro-inflammatory responses observed in immune cells upon CpG stimulation, LGG^®^ genomic DNA and ODN had an anti-inflammatory effect in IECs. TLR9 is expressed both apically and basolaterally in polarized IECs, but evoke quite different signaling responses, thereby controlling tolerance and inflammation [153]. In HT-29 and T84 polarized cell monolayers, TNF-α induced NF-κB activation and IL-8 expression. Apically applied LGG^®^ genomic DNA and ODN, activating apical TLR9, attenuated TNF-α-induced NF-κB activation by reducing I_Κ_B_α_ degradation and p38 phosphorylation. Importantly, bacterial genomic DNA was detected exclusively on the apical side, with no access to basolateral TLR9, preventing the activation of the inflammatory NF-κB pathway [154]. Consistent with these findings, oral priming with ID35-containing carbonate apatite-based particles (ID35caps) ameliorated symptoms and inflammatory colonic injury in a mouse DSS-colitis model. DNA was protected from acid degradation by ID35caps and improved body weight change, fecal bleeding, and stool consistency in DSS-treated mice [155].

## 8. Comparison of LGG^®^ Effector Molecules and Overall Effects

The diverse responses to the LGG^®^ effector molecules in epithelial and immune cells underscore the overall complexity and regulated impact of LGG^®^ on the gut mucosa and immune system. Two studies have undertaken a comparative analysis of the effects of LGG^®^-derived molecules, both individually and in combination.

In one study, the stimulation of confluent IEC monolayers with LPS triggered a pronounced pro-inflammatory response, marked by a substantial increase in TLR4 expression, and to a lesser extent, TLR2 and TLR9. This resulted in the secretion of pro-inflammatory cytokines (TNF-α, IL-6, and IL-12). LGG^®^ treatment of IEC monolayers also triggered a pro-inflammatory response, albeit at significantly lower levels than LPS, and without activating TLR4. Pre-treatment of monolayers with LGG^®^, before LPS stimulation, attenuated the LPS-induced inflammatory response significantly by reducing TLR2, and TLR9 activation, as well as cytokine expression. Similar effects were observed when monolayers were pre-treated with purified LGG^®^ surface layer proteins (SLP) or EPS. Pre-treatment with genomic LGG^®^ DNA had no effect on pro-inflammatory markers, whereas the CpG-motif, ID35, augmented TLR9, TNF-α, and IL-12 expression. Intact LGG^®^ cells, SLP, and EPS attenuated LPS-induced MAPK and NF-κB signaling, while ID35 activated NF-κB signaling. In essence, LGG^®^ demonstrated a capacity to alleviate inflammation in LPS-stimulated IECs by modulating TLRs and inhibiting MAPK and NF-κB signaling [125].

Similarly, the anti-inflammatory effects of LGG^®^ effector molecules on LPS-stimulated pro-inflammatory responses were observed in macrophages. Pre-treatment of macrophages with SLP, genomic DNA, or SLP and genomic DNA before LPS treatment significantly suppressed the gene expression of TLR2, TLR4 and TLR9 and pro-inflammatory cytokines (TNF-α, IL-6). ID35 augmented TNF-α and IL-6 expression in LPS-induced macrophages, while SLP, in combination with ID35, reduced the ID35-induced cytokine and TLR9 expression. The anti-inflammatory effects of LGG^®^ effector molecules in macrophages were achieved through the inhibition of TLRs, MAPK, and NF-κB activation [156].

## 9. Conclusions and Future Perspectives

Collectively, the majority of LGG^®^ effector molecules have regulatory functions on the intestinal epithelium and the innate immune system and promote intestinal homeostasis and immune tolerance (Figure 1 and Figure 2). Very strong protective effects have been found of the secreted proteins (p40, p75, and HM0539), which are also present on the LGG^®^ cell wall, and on extracellular vesicles. Intracellular signaling networks are set in motion through EGFR activation, leading to anti-apoptosis and cell protection of the epithelium, preservation of cellular junctions, and increased mucus production. Furthermore, these proteins contribute to the enhancement of immune tolerance and homeostasis through various mechanisms, including the reduction in TLR4/MyD88/NF-κB pro-inflammatory signaling, stimulation of regulatory T-cell populations via TGF-β, and the augmentation of B cell IgA class switching and IgA production. The EPS derived from LGG^®^ exhibit a potent inhibitory effect on p38/MAPK and NF-κB signaling, attenuating pro-inflammatory responses and displaying protective anti-oxidative effects. LTA from LGG^®^ induces T-helper 1 immune responses through TLR2/6-activated MyD88/NF-κB signaling, while also demonstrating the capability to suppress exaggerated inflammatory responses. Additionally, LTA contributes to an increased number of IgA-positive B cells. Unmethylated CpG-rich DNA motifs from LGG^®^ stimulate TLR9 activated MyD88/NF-κB signaling leading to a T-helper 1 immune response, B cell proliferation, and suppression of antigen-specific IgE production. Interestingly, in the intestinal epithelium, CpG motifs and LGG^®^ genomic DNA manifest anti-inflammatory and protective effects. Purified pili, covering the cell surface of LGG^®^, display both pro- and anti-inflammatory effects. However, the most important function of pili is to ensure close contact between LGG^®^ and host cells, facilitating robust interactions with other potent effector molecules.

Over the past two decades, a wealth of pre-clinical research on LGG^®^ has provided a profound and unique insight into how this probiotic strain may influence human health at the molecular level. This extensive body of work has unveiled effector molecules and elucidated their interactions with fundamental physiological mechanisms. The implications of this knowledge should be considered in the interpretation of outcomes of LGG^®^ clinical trials and in the design of new trials. Further trials are needed to ascertain whether the mechanisms identified in pre-clinical studies are also effective in humans and what their relevance is to health and well-being. Additionally, this knowledge may advise biological markers of LGG^®^ effectiveness, facilitating the translation of pre-clinical research into tangible human health benefits. The insights gained from research on LGG^®^ should inspire and guide similar endeavors with other probiotic strains. Lastly, the uncovered mechanisms and involved molecules offer promising avenues for establishing new targets in quality control and proposing optimization parameters for the industrial production and commercial formulations of LGG^®^.

## Figures and Tables

**Figure 1 microorganisms-12-00794-f001:**
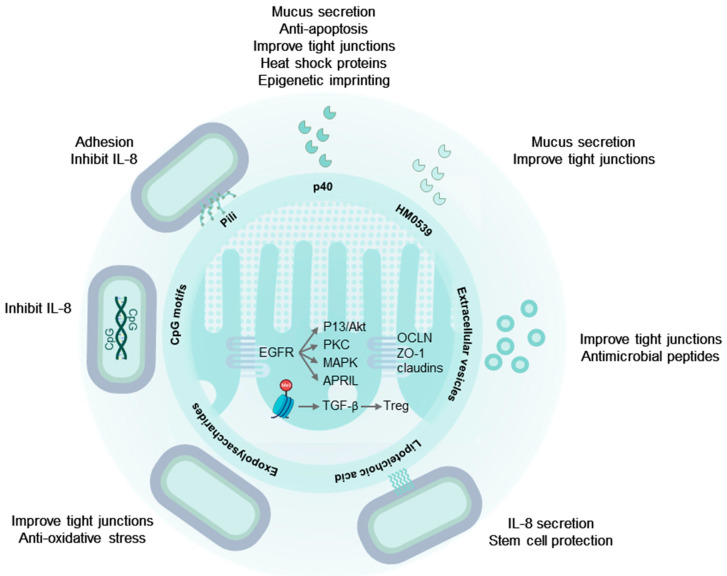
LGG^®^ effector molecules protect the intestinal epithelium. LGG^®^ pili ensure colonization and close contact with the epithelium. Secreted and cell wall associated p40 proteins activate EGFR and downstream signaling, promoting mucus secretion, cell survival, and maintaining cellular tight and adherens junctions. Additionally, p40 induces mucosal epigenetic imprinting, resulting in sustained TGF-β secretion by epithelial cells, driving regulatory T-cell differentiation. EPS from LGG^®^ protect tight junctions and reduce oxidative stress via the Nrf2-Keap1-ARE signaling pathway, while LTA directs the migration of protective prostaglandin-expressing mesenchymal stem cells to areas near epithelial stem cells. Extracellular vesicles mirror these effects through their cargo of LGG^®^-derived molecules. Tryptophan metabolites within extracellular vesicles activate the aryl hydrocarbon receptor, preserve tight junctions, and promote IL-22 secretion in lymphocytes. This, in turn, stimulates the release of antimicrobial peptides from Paneth cells. See text for abbreviations.

**Figure 2 microorganisms-12-00794-f002:**
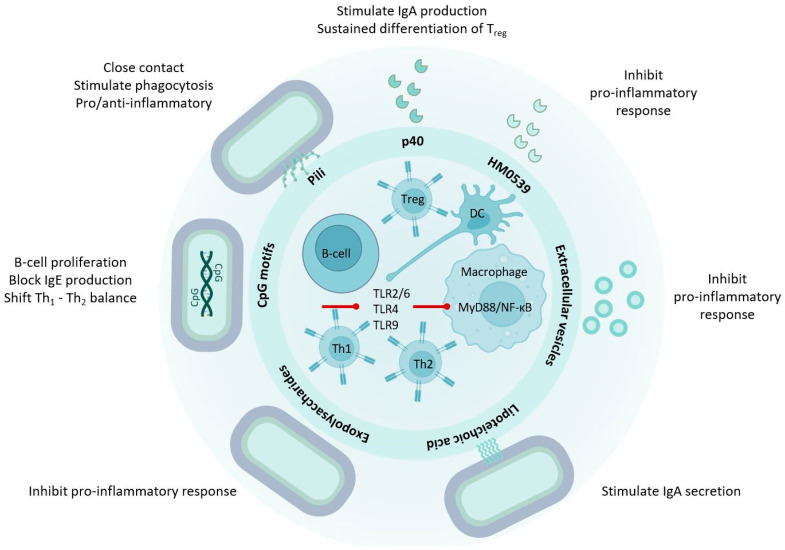
LGG^®^ effector molecules contribute to immune homeostasis. LGG^®^ effector molecules stimulate immune regulation and tolerance through interactions with macrophages, lymphocytes, antigen presenting cells, and intestinal epithelial cells. Various LGG^®^-derived molecules, such as p40, HM0539, and EPS, mitigate inflammation induced by LPS or bacterial pathogens by modulating TLR expression and NF-κB activation. Conversely, D-alanyl ester-substituted LTA from LGG^®^ activates TLR2/6 and NF-κB signaling in dendritic cells. The EGFR-stimulated APRIL expression in epithelial cells, induced by p40, initiates B cell IgA class switching in the lamina propria, leading to increased IgA production. Additionally, LTA contributes to an elevated number of IgA-positive B cells. The heightened expression of TGF-β in epithelial cells induced by p40 plays a crucial role in sustaining the differentiation of regulatory T-cells, which are essential for mucosal immune homeostasis. Unmethylated CpG-rich DNA motifs stimulate dendritic cells through TLR9 and NF-κB activation, promoting T-helper 1 and inhibiting T-helper 2 cell differentiation. See text for abbreviations.

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
