# Peer review of "Molecular Mechanisms of Lacticaseibacillus rhamnosus, LGG® Probiotic Function"

_microorganisms, 2024, doi:10.3390/microorganisms12040794_

Round 1

Reviewer 1 Report

Comments and Suggestions for Authors

The manuscript detailed reviewing the beneficial effects of Lacticaseibacillus rhamnosus LGG on human health, the probiotic effector molecules derived from LGG, and the regulatory effects and mechanisms on human gut.  In general, the manuscript is well written and the  English language is fine.

Author Response

This reviewer had no suggestions for revision

Reviewer 2 Report

Comments and Suggestions for Authors

This manuscript reviews the existing knowledge on the molecular interactions between LGG® and the host. As the most studied probiotic, LGG's effector molecules, including secreted proteins, surface-anchored proteins, polysaccharides, and lipoteichoic acids, and their interaction with host physiology were highlighted. Please note the following question.

Line 21 keywords: Lacticaseibacillus rhamnosus should be italicized.

In the introduction, please highlight how this review fills a gap in the existing literature. Have similar papers been published? What databases are the references for this review from and what is the time frame?

Can you cite more literature from the last 3 years to support the argument of the review?

Author Response

Line 21; Lacticaseibacillus rhamnosus italicized.

The reviewer requests a justification of the paper. We think the paper is relevant as it is an update of our understanding of LGG molecular mechanisms of action since the latest review from 2014 by Segers & Lebeer. 89 references published since 2014 have been included. We added this line to the introduction: “Significant scientific contributions have been made since the review by Segers & Lebeer was published ten years ago”.

The sources of references for this review is our own collection of papers as well as PubMed searches. The review is not intented to be a systematic review, but is a thorough review of papers we have found relevant to the topic. Some papers were not included because they did not contribute significantly new understanding of the effector molecules, but basically repeated what was already shown.

To the reviewer’s wish for more recent papers, we argue that important discoveries of LGG effector molecules were actually done years ago. However, 27 referenced papers are from the last 3 years. We have not found additional papers in this time frame, that improve our understanding of LGG effector molecules.

Reviewer 3 Report

Comments and Suggestions for Authors

This review is elegantly written and exhustively desrcibes all the aspects of L. rhamnosus used as probiotic. I would like to stimulate the authors to add some comments related to the difficulty to compare biological data obtained by different preparation/isolation/purification techniques when working with the same strain or species. Small changes in the conditions/procedures must have an impact on chemical composition of the OMVs as an example. Indeed, working in this field, I have also noted a limited stability of this material (OMVs) during time. Consequently, the results related to this and other components or derived products from probioitics can achieve or not the same outcome.

Author Response

We understand the reviewers concern about the potential influence of different preparation/isolation/purification methods on probiotic effector molecules.

We have already (partly) addressed this concern in the manuscript in lines 179-186. We have now expanded this paragraph by including a sentence on the potential divergences introduced by culturing conditions and added three references that illustrate such effects. “To further complicate this, bacterial synthesis of structural and secreted compounds may be sensitive to fermentation conditions such as specific nutrient availability or temperature[72–74]. Consequently, findings from such studies should be approached with caution, acknowledging the potential influence of growth conditions and impurities on the results.”